# Style Over Substance: Evaluation Biases for Large Language Models

## Abstract

As large language models (LLMs) continue to advance, accurately and comprehensively evaluating their performance becomes increasingly challenging. Human evaluations are conventionally considered the gold standard in natural language generation, but recent advancements incorporate state-of-the-art LLMs as proxies for human judges in evaluation processes. However, the extent to which humans and LLMs are capable evaluators remains uncertain. This study investigates the behavior of crowd-sourced and expert annotators, as well as LLMs, when comparing outputs from different models. To achieve this, we curate a dataset of intentionally flawed machine-generated answers. Our findings reveal a concerning bias in the evaluation process, as answers with factual errors are rated more favorably than answers that are too short or contained grammatical errors. To address this issue, we propose independently evaluating machine-generated text across multiple dimensions, rather than merging all the evaluation aspects into a single score. We instantiate this idea with the Elo rating system, resulting in the Multi-Elo Rating System (MERS). Empirical results from our study reveal that this proposed approach significantly enhances the quality of LLM-based evaluations, particularly in terms of factual accuracy. However, there is no significant improvement in crowd-sourced-based evaluations, indicating the need for further investigation and refinement.

## 1 Introduction

Recent advancements in the field of natural language processing have demonstrated that the utilization of supervised instruction fine-tuning and reinforcement learning from human feedback (RLHF) can yield substantial improvements in the performance of large language models (LLMs) with respect to their ability to comprehend and execute instructions (Ouyang et al., 2022; Wei et al., 2022; Sanh et al., 2022; Chung et al., 2022; OpenAI, 2023; Wu et al., 2023; Li et al., 2023a; Lyu et al., 2023). This progress signifies a significant stride in the domain of language model development. However, the assessment of these enhanced LLMs presents a notable challenge, particularly when confronted with more generalized instructions that entail open-ended responses. Such instructions often lack a definitive metric for evaluation within the confines of traditional natural language processing benchmarks. In response to this challenge, several recent studies have introduced a novel approach wherein either human or LLM judges are enlisted to adjudicate between two LLM-generated outputs (Askell et al., 2021; Bai et al., 2022a; Srivastava et al., 2022; Chiang et al., 2023; Dettmers et al., 2023; Zheng et al., 2023). This evaluation method enables the computation of an Elo rating, a ranking system originally devised for chess (Elo, 1967). Nonetheless, an important question arises concerning the qualifications of human and LLM judges to serve as effective evaluators in this context. Evaluating model outputs encompasses a multifaceted decision-making process, and it remains an open question whether these judges possess the requisite expertise to accurately determine the superior model output. Further research is needed to address this inquiry comprehensively and refine the evaluation procedures for enhanced LLMs.

In this study, we systematically generate a set of responses, considering factors such as language proficiency, factual accuracy, and response length. We employ 40 general-purpose questions sourced from (Chiang et al., 2023) that do not require specialized expertise to ensure the generalization of our study and reduce annotation difficulty. The answers for these questions are generated by GPT-4 with

| | Answer Features | | | Elo Ratings | | | |
| --- | --- | --- | --- | --- | --- | --- | --- |
| | # of words | Language Errors | # of Factual Errors | Human | | GPT-4 | Claude-1 |
| | | | | Crowd | Expert | | |
| `Correct` | ≈ 100 | N.A. | 0 | 1091 | 1162 | 1482 | 1320 |
| `  + Short` | ≈ 50 | N.A. | 0 | 970 | 1029 | 1096 | 1052 |
| `One Minor Factual Error` | ≈ 100 | N.A. | 1, minor | 1074 | 1137 | 1415 | 1265 |
| `  + Short` | ≈ 50 | N.A. | 1, minor | 1002 | 964 | 988 | 997 |
| `Several Minor Factual Errors` | ≈ 100 | N.A. | ≈ 3, minor | 1032 | 1024 | 1206 | 1182 |
| `  + Short` | ≈ 50 | N.A. | ≈ 3, minor | 952 | 873 | 851 | 891 |
| `Several Major Factual Errors` | ≈ 100 | N.A. | ≈ 3, major | 1025 | 892 | 861 | 979 |
| `  + Short` | ≈ 50 | N.A. | ≈ 3, major | 937 | 832 | 710 | 782 |
| `Advanced Learner` | ≈ 100 | Spelling | 0 | 1041 | 1138 | 1213 | 1126 |
| `  + Short` | ≈ 50 | Spelling | 0 | 941 | 986 | 824 | 841 |
| `Intermediate Learner` | ≈ 100 | Grammatical | 0 | 1015 | 1108 | 771 | 904 |
| `  + Short` | ≈ 50 | Grammatical | 0 | 921 | 855 | 582 | 662 |

Table 1: Elo ratings for answers in different settings based on the annotations given by crowd-sourced annotators, expert annotators, GPT-4, and Claude-1.

specific instructions. To probe the potential impact of language proficiency towards human and LLM judgments, we instruct GPT-4 to emulate an advanced English learner, occasionally incorporating spelling errors, or an intermediate English learner, occasionally introducing grammatical mistakes during the response generation process. To probe factual accuracy, we direct GPT-4 to include varying degrees of factual errors in the responses. Lastly, we explore the influence of response length by instructing GPT-4 to generate answers of differing lengths. To ensure that the generated responses conformed to the desired criteria, we conduct manual reviews and carry out post-editing as necessary. Subsequently, after obtaining the collection of responses, we conduct annotations with a diverse pool of annotators, including crowd-sourced annotators, expert annotators, and LLMs. We then calculate the Elo ratings for each "model" based on their judgments to understand their preferences when deciding on high-quality model outputs.[1]

In the course of our investigation (Table 1 and Section 3), we uncover intriguing findings related to the assessment of answer quality. Firstly, we observe hesitancy among humans, including both crowd-sourced and expert annotators, in determining answer quality, while LLMs exhibit greater certainty in their evaluations. Furthermore, we notice a substantial difference in fact-checking capabilities between human judges and LLMs. Human judges generally do not thoroughly fact-check answers unless the factual error is glaringly evident, whereas LLMs demonstrate some degree of fact-checking ability, albeit with imperfections. Another significant finding is that both human judges and LLMs tend to favor longer text when evaluating answers. Interestingly, crowd-sourced annotators appear to be (almost) unbiased in their evaluation of answer ordering, while LLMs lean toward the first answer and expert annotators prefer the second answer.

We note that a single unified measure to evaluate LLM is not sufficient, considering that many aspects play a role when determining its quality. Moreover, some aspects are arguably more important than others (e.g., factuality). Yet, we see judges preferring factually incorrect models over grammatically incorrect or short ones. With these identified issues, we present a Multi-Elo Rating System designed to assess machine-generated text from multiple dimensions.

Our proposed approach requires human judges and LLMs to evaluate the machine-generated text independently from three aspects: "Accuracy", "Helpfulness", and "Language". This allows us to achieve a more comprehensive and transparent understanding of the quality of machine-generated text. Our empirical findings demonstrate a significant improvement in the evaluation quality of GPT-4, particularly in terms of factual accuracy. However, we observe that humans still exhibit indecisiveness in their assessments.

---

[1]In this work, we refer to different settings in answering questions as different models, although all the answers are generated by GPT-4.

Based on our findings, we highly recommend that practitioners evaluate machine-generated text from various perspectives rather than depending solely on a single unified measure. Additionally, we advise practitioners to exercise caution when using crowd-sourced human annotators to assess the performance of LLMs.

## 2 EVALUATION METHOD

In this section, we describe our evaluation approach. We introduce the Elo rating system and the process of generating incorrect answers for each model. We also discuss the human and LLM evaluation methods utilized.

### 2.1 ELO RATING SYSTEM

The Elo rating system is a method used to calculate the relative skill levels of players in two-player games, such as chess. Given two players $\mathcal{A}$ and $\mathcal{B}$ whose Elo ratings are $\mathcal{R}_A$ and $\mathcal{R}_B$ respectively, the expected score for these two players are:

$$\mathcal{E}_A = \frac{1}{1 + 10^{\frac{\mathcal{R}_B - \mathcal{R}_A}{400}}}, \text{and } \mathcal{E}_B = \frac{1}{1 + 10^{\frac{\mathcal{R}_A - \mathcal{R}_B}{400}}}. \tag{1}$$

Suppose the player $\mathcal{A}$ is expect to obtain $\mathcal{E}_A$ scores from this game but actually get $\mathcal{S}_A$ scores, the updated Elo rating of player $\mathcal{A}$ is:

$$\mathcal{R}'_A = \mathcal{R}_A + \mathcal{K} \cdot (\mathcal{S}_A - \mathcal{E}_A), \tag{2}$$

where $\mathcal{K}$ is adjustment parameter, called the $\mathcal{K}$-factor. Equation 2 is also used for obtaining $\mathcal{R}'_B$. Following Chiang et al. (2023), we set $\mathcal{K} = 32$. Additionally, if $\mathcal{A}$ is better than $\mathcal{B}$, we set $\mathcal{S}_A = 1$ and $\mathcal{S}_B = 0$. If $\mathcal{B}$ is better than $\mathcal{A}$, we set $\mathcal{S}_A = 0$ and $\mathcal{S}_B = 1$. We set both $\mathcal{S}_A = 0.5$ and $\mathcal{S}_B = 0.5$ if both players are equally good. Moreover, to minimize the influence of the ordering of games, the Elo rating calculation is performed 10,000 times with varying random orderings as suggested by Dettmers et al. (2023).

### 2.2 ANSWER GENERATION

Chiang et al. (2023) release a set of 80 questions categorized into 8 categories to evaluate the generation capability of LLMs. However, due to the requirement for specialized expertise to answer some of these questions, we exclude the "fermi", "coding", and "math" questions, as they typically demand extra efforts to evaluate the answers. This step aims to reduce the potential impact of varying human raters' capabilities on the evaluation process. Additionally, we also remove the "roleplay" and "writing" questions, as they involve creative writing and are prone to subjectivity in human assessment. As a result, our final question set consists of 40 questions, focusing on the "generic", "knowledge", "common sense", and "counterfactual" categories.

Once we have the set of questions, we require GPT-4 to generate answers with specific error types in addition to providing the correct answers. We provide some examples of these error types in Table 2. Regarding language quality errors, we ask GPT-4 to respond as either an advanced English learner or an intermediate English learner. The answers generated by an advanced English learner may occasionally contain spelling errors, while those from an intermediate English learner commonly include grammatical mistakes. In terms of factual accuracy, we expect GPT-4 to produce answers with either minor or major errors. Minor errors primarily involve fabricated names or incorrect numbers, while major errors contain incorrect facts and suggestions. Furthermore, we utilize GPT-4 to generate both long (approximately 100 words) and short (approximately 50 words) answers for each question to investigate the preference of both humans and LLMs regarding answer length. Hence, there are 12 models (settings) in total in this study. We present the prompts used for answer generation in Appendix A. To ensure unbiased evaluation results regarding answer ordering, all evaluators, including crowd-sourced human annotators, expert annotators, and LLMs, evaluate all answer pairs from both forward and reversed directions. In the end, we have 5280 unique pairwise comparisons across all 12 models and 40 questions.

During the generation process, we have noticed that GPT-4 may not always fulfill our requirements. For instance, the output sometime may not contain any factual or language errors. Therefore, we manually review all the answers and make necessary edits to ensure they align with our requirements.

| Model type | Output |
|---|---|
| Correct | Effective ways to deal with stress include regular exercise, maintaining a healthy diet, getting enough sleep, and taking time for relaxation activities like meditation. It's also important to have a support system in place, whether that's friends, family, or a mental health professional. |
| Minor Errors | Effective ways to deal with stress include physical activities like playing tennis, doing yoga, or running a marathon. Also, adopt healthy habits like sleeping for 10 hours, eating three full meals a day, and avoiding caffeine. Lastly, meditation, guided by a professional like John Zen, can be very soothing. |
| Major Errors | Some effective ways to deal with stress include drinking 10 cups of coffee a day, binge-watching Netflix for at least 5 hours, and attending Dr. Monty Farquhar's stress-relief seminars. Additionally, practicing the art of Floga, a combination of Flute-playing and Yoga, can significantly reduce stress levels. |
| Advanced Learner | Whenevr feeling stressed, always remember to take deep breathes. This can calm nerves. Regular physical exercise also reduce stress hormonal levels. A balance diet promoting overral well-being can lead to up healty stress response. Finally, finding hobby or activity you enjoy can help distract you from stressful situations. |
| Intermediate Learner | Deal with stress, you can do exercise regularly, practicing a meditation, get with plenty sleep, and eat healthy foods also. You can too to connect with others so express your feelings, and avoiding caffeine, alcohol, and nicotine, and take time to relax and have fun. |

Table 2: Examples of different error types for the question "What are the most effective ways to deal with stress?". Factual errors are highlighted in red and language errors are highlighted in orange.

## 2.3 CROWD-SOURCED EVALUATION

Crowd-sourced human annotators are commonly used to assess the quality of machine-generated text. For this study, we utilize Amazon Mechanical Turk (AMT) to collect text evaluations from human raters in NLG evaluations. To minimize the potential impact of annotator demographics, we only recruit crowd-sourced human annotators from the United States. We also ensure the quality of annotations by exclusively offering tasks to annotators who have completed over 5,000 assignments and maintain a success rate of 99% or higher. To ensure the representativeness of the annotators, we limit each annotator to provide no more than 20 annotations and enforce a strict maximum of 50 annotations per person. As a result, we engage 377 unique annotators, each contributing an average of 14 annotations.

During annotation, we provide a simple interface that mainly follows the guidelines of Dettmers et al. (2023), as shown in Appendix C. Annotators are presented with a question and two model-generated responses placed side-by-side. Their task is to select the better output or indicate a tie between them. To ensure the annotators' attentiveness and thorough reading of the responses, we incorporate a mandatory 20-second delay before they can submit their answers. Furthermore, we anonymize the model name from our human annotators.

## 2.4 EXPERT EVALUATION

To address concerns about the reliability of crowd-sourced annotators, we have implemented a parallel system involving a team of 20 expert annotators. Each member of this dedicated team holds at least a master's degree from an English-speaking country, ensuring a comprehensive understanding of the language and proficiency in annotation nuances. It is important to note that employing expert annotators comes at a higher cost than using crowd-sourced alternatives. Therefore, we sample 200 games out of a total pool of 5280 games, which is statistically sufficient to yield meaningful insights into the model's performance.[2] To ensure a fair comparison, we provide the same annotation instructions to both the expert team and the crowd-sourced participants. Each expert annotator is assigned to annotate 10 games, taking them approximately 20 minutes. This approach enables us to accurately evaluate the effectiveness and reliability of each annotation method.

## 2.5 LLM EVALUATION

As human evaluation can be costly and inefficient, there is an increase in the use of advanced LLMs, such as GPT-4, to evaluate model outputs. In our work, we also use LLMs as judges to assess answer

---

[2] The US Chess Federation believes that 25 games are enough to assess a player's ability, and in our sample of 200 games, each model participated in at least 28 games. Source: `https://new.uschess.org/frequently-asked-questions-member-services-area`

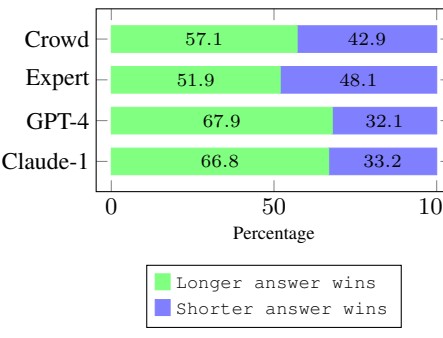
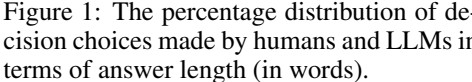
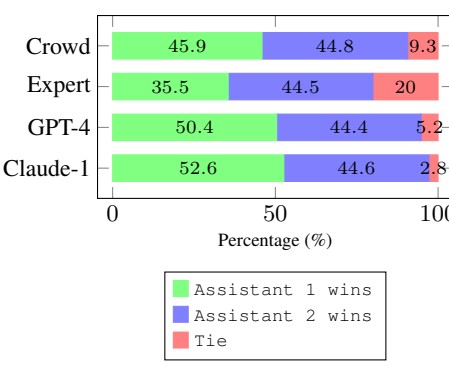

Figure 1: The percentage distribution of decision choices made by humans and LLMs in terms of answer length (in words).

Figure 2: The distribution of decision choices made by humans and LLMs.

quality. However, previous studies relies solely on GPT-4 as the LLM judge (Chiang et al., 2023; Li et al., 2023a; Zheng et al., 2023), which may not be appropriate for our work as our answers are refined by humans after being generated by GPT-4. This raises concerns about potential biases that GPT-4 may have towards its own outputs, which could skew the evaluation process. To ensure fair comparisons and mitigate any bias, we introduce Claude-1 from Anthropic (Bai et al., 2022b) as an additional LLM judge, in addition to GPT-4.[3] By incorporating multiple LLM judges, we can establish a more comprehensive and unbiased assessment of the generated answers in our study.

We utilize the evaluation prompt from Dettmers et al. (2023), as presented in Appendix B. The prompt assesses the answers based on their helpfulness, relevance, accuracy, and level of detail, while also aiming to avoid bias related to answer ordering.

## 3 ANALYSIS

In this study, we assess the performance of 12 models using Elo ratings, as evaluated by crowd-sourced annotators, expert annotators, GPT-4, and Claude-1. The Elo ratings are presented in Table 1. As anticipated, the standard correct model attains the highest Elo score across all human judges. Nevertheless, intriguing disparities emerge when comparing the Elo results from human judges to those generated by GPT-4 and Claude-1. In this section, we delve into a comprehensive analysis of these distinctions.

**All the judges exhibit a bias toward longer texts.** Text evaluation by both human and LLM judges often exhibits a bias towards longer responses, where GPT-4 demonstrates the most bias and the expert annotators demonstrate the least bias, as depicted in Figure 1. This inclination is expected since one of the scoring criteria requested is "the level of detail", which often correlates with the length of the output. However, such an inclination is arguably undesirable. For example, we observe that GPT-4 considers "Several Minor Factual Errors" (1206 Elo) to be better than "Correct + Short" (1096 Elo). When manually reviewing the justifications provided by GPT-4 for its evaluation decisions, we observe that GPT-4 sometimes considers the response of "Several Major Factual Errors" as superior to that of "Correct + Short", even when factual errors are detected.

**Humans are indecisive.** In contrast to LLM judges, both expert and crowd-sourced annotators tend to demonstrate indecisiveness, leading to Elo scores that remain relatively close to the initial value of 1000. The Elo scores assigned by crowd-sourced annotators range from 926 to 1086, while those by expert annotators range from 832 to 1162. These ranges are significantly narrower than those observed for GPT-4 and Claude-1. Interestingly, human judges are more inclined to assign "tie" judgments compared to LLMs, as illustrated in Figure 2. These findings raise important questions about the level of reading comprehension and attention to detail demonstrated by human judges, especially when they are crowd-sourced workers evaluating answers.

---

[3]GPT-4 signature: `gpt-4-0613` and Claude-1 signature: `Claude-1.3`

**The order of answers affects the judges' decisions.** Our analysis also reveals an interesting bias toward the order of answers, which is evident in the assessments made by both human judges and LLM judges. As illustrated in Figure 2, it can be observed that the crowd-sourced annotators, GPT-4, and Claude-1 exhibit a distinct and notable preference for Assistant 1. This intriguing finding is consistent with prior research (Wang et al., 2023), further underscoring the significance of this bias. Interestingly, the expert annotators favor the answer given by Assistant 2 and are more likely to assign "tie" judgments. To ensure fairness and comprehensive evaluation, we strongly recommend considering both answer orders when analyzing the performance of the systems. We leave the further study of the behavior of human annotators for future work.

**Crowd-sourced annotators lack fact-checking, while experts and LLM judges can fact-check, albeit imperfectly.** The close Elo scores of those model with factual errors from crowd-sourced annotators in Table 1 suggest inconsistent and inadequate fact-checking of generated content, raising concerns about false information and malicious intent in LLM-generated outputs. People are vulnerable to believing such outputs, particularly when they appear convincing. Although expert annotators are more diligent in fact-checking, the general public's behavior towards LLM output tends to be closer to that of crowd-sourced annotators, posing significant safety risks. On the other hand, LLM judges do notice errors, but not consistently. When the LLM fails to detect inaccuracies, it often favors flawed outputs over shorter or grammatically imprecise responses.

**LLMs reach consensus, while humans often do not.** We assess inter-annotator agreement using Cohen's kappa coefficient $\kappa$ (Cohen, 1960) and present the results in Table 3. Our analysis, following the interpretation of $\kappa$ by McHugh (2012), shows that only GPT-4 and Claude-1 achieve moderate agreement, while other comparisons demonstrate only slight agreement. This confirms that human annotators exhibit inconsistencies during annotation. Zheng et al. (2023) define the agreement between two types of judges as the

|          | Crowd | Expert | GPT-4 | Claude-1 |
|----------|-------|--------|-------|----------|
| Crowd    | —     | 0.08   | 0.11  | 0.10     |
| Expert   | 0.08  | —      | 0.09  | 0.14     |
| GPT-4    | 0.11  | 0.09   | —     | 0.51     |
| Claude-1 | 0.10  | 0.14   | 0.51  | —        |

Table 3: Pairwise inter-annotator agreement measured by Cohen's kappa coefficient $\kappa$. The pairs involving the experts only cover 200 games.

probability of non-identical individuals of each type agreeing on a randomly selected question and shows an approximately 80% agreement between LLMs and crowd-sourced humans, which seems to contradict our findings. However, this discrepancy arises because they remove ties and inconsistent annotations, while we consider all annotations. When ties and inconsistencies are retained, Zheng et al. (2023) report an approximately 60% agreement between LLMs and crowd-sourced humans, which is slightly better than random guessing and aligns with our findings.

## 4 MULTI-ELO RATING SYSTEM

As discussed in Section 3, we identify the limitations in the evaluation of machine-generated answers based on human and LLM judges. We observe that the current widely used evaluation approach fails to yield satisfactory results and lacks a comprehensive understanding of the factors influencing the outcomes. Collapsing multiple decision components (e.g., accuracy, level of details, relevance, language, helpfulness, etc.) into a single score undermines the importance of individual components and introduces subjectivity regarding their relative significance in the final decision.

To overcome these limitations, we propose a novel multi-dimensional evaluation approach for assessing the outputs of LLMs, which we refer to as the Multi-Elo Rating System (MERS). This approach is inspired by machine translation research, where evaluations are often divided into at least two parts: fluency and adequacy. Recent advancements in MT evaluation also advocate breaking down the evaluation into multiple aspects (Lommel et al., 2014). In this section, we present a detailed description of our evaluation methodology and present the results we obtained through this new approach.

| | Accuracy | | | Helpfulness | | | Language | | |
|---|---|---|---|---|---|---|---|---|---|
| | Crowd | Expert | GPT-4 | Crowd | Expert | GPT-4 | Crowd | Expert | GPT-4 |
| C. | 1056 | 1180 | 1200 | 1045 | 1208 | 1384 | 1036 | 1109 | 1415 |
| +S. | 963 | 1040 | 1158 | 983 | 979 | 1009 | 1007 | 1068 | 1199 |
| OMin.FE | 1026 | 1090 | 1120 | 1048 | 1153 | 1378 | 1019 | 1114 | 1334 |
| +S. | 978 | 898 | 1016 | 993 | 941 | 965 | 990 | 1012 | 1109 |
| SMin.F | 1036 | 1044 | 993 | 1051 | 1069 | 1248 | 1029 | 1096 | 1200 |
| +S. | 978 | 931 | 857 | 956 | 865 | 845 | 996 | 935 | 988 |
| SMaj.FE | 1030 | 963 | 794 | 1037 | 1015 | 926 | 1023 | 1010 | 995 |
| +S. | 955 | 787 | 746 | 940 | 766 | 726 | 982 | 879 | 871 |
| AL | 1028 | 1121 | 1139 | 1032 | 1146 | 1196 | 1004 | 1039 | 1051 |
| +S. | 979 | 971 | 1051 | 969 | 891 | 804 | 994 | 863 | 814 |
| IL | 1015 | 1076 | 1018 | 1002 | 1095 | 908 | 991 | 992 | 560 |
| +S. | 956 | 898 | 908 | 945 | 872 | 612 | 930 | 884 | 465 |

Table 4: Elo ratings for different models with regard to "Accuracy", "Helpfulness", and "Language" given by crowd-sourced annotators, expert annotators, and GPT-4. `C.` stands for `Correct`. `OMin.FE` stands for `One Minor Factual Error`. `SMin.F` stands for `Several Minor Factual Errors`. `SMaj.FE` stands for `Several Major Factual Errors`. `AL` stands for `Advanced Learner`. `IL` stands for `Intermediate Learner`. `S.` stands for `Short`.

## 4.1 METHODOLOGY

The Multidimensional Quality Metrics (MQM) framework provides a comprehensive approach for evaluating and establishing standards for translation quality (Lommel et al., 2014). Drawing inspiration from this framework, we propose a similar approach to evaluate the outputs of LLMs from multiple dimensions.

Our evaluation focuses on three main dimensions of the generated text, as outlined below:

- **Accuracy**: The accuracy of the text involves considering factual correctness and logical consistency.
- **Helpfulness**: The helpfulness of the text involves considering its relevance of the information and whether it addresses the question given, taking into account the depth of the response given.
- **Language**: The language of the text involves considering its clarity, coherence, grammar, syntax, and tone.

The quality of an answer is dependent on its specific context. For instance, if a model gives a detailed but complicated explanation of black holes to an 8-year-old, the answer may be accurate but not useful. Conversely, if a model is asked to compose an email and produces a message with incorrect information, the response may lack accuracy but still have some value. By taking this multi-dimensional approach, we can gain a clearer understanding of model performance and prioritize different aspects based on our individual requirements.

To facilitate the multi-dimensional evaluation by human annotators, we introduce a simple modification to the interface, asking them to rate the quality across the three different aspects, as shown in Appendix C. Additionally, we experiment with two approaches for GPT-4 evaluation: asking three independent queries versus a single query that requests judgments for all three aspects together. In this paper, we report the results obtained from asking three independent queries. More details can be found at Appendix D.

## 4.2 STOP USING CROWD-SOURCED ANNOTATORS!

In this section, we compare the annotation outcomes provided by both crowd-sourced and expert annotators, as presented in Table 4. We examine the aspects of "Accuracy", "Helpfulness", and "Language" in the evaluation. Regarding "Accuracy", we find that expert annotators are proficient in identifying factual errors in answers, although not entirely flawless. However, crowd-sourced annotators exhibit indecisiveness in their evaluations. Notably, the crowd-sourced annotators per-

ceive `Several Major Factual Errors + Short` (955 Elo) and `Correct + Short` (963 Elo) as nearly equally good. Regarding "Helpfulness", the expert annotators display a stronger preference for longer answers, while the crowd-sourced annotators only slightly favor them, as evidenced by their Elo scores. In terms of "Language", both expert and crowd-sourced annotators face challenges in recognizing spelling or grammatical errors, suggesting that humans may be less sensitive to language errors and aligning with the findings of Clark et al. (2021).

In conclusion, the expert annotators outperform the crowd-sourced annotators in the evaluation, despite not being entirely error-free themselves. These results serve as a warning against overreliance on crowd-sourced judgments and also highlight concerns regarding the general audience's ability to critically interpret LLM's output.

## 4.3 EXPERTS VERSUS GPT-4

In this section, we discuss the difference between the expert annotators and GPT-4 in evaluation from multiple dimensions and present the results in Table 4. Regarding the "Accuracy" dimension, it is noteworthy that the Elo scores for factual accuracy closely align with the single Elo scores presented in Table 1, suggesting that expert annotators indeed prioritize factual accuracy during evaluation. GPT-4 can also effectively rank models based on the severity of the errors. Regarding the "Helpfulness" dimension, both expert annotators and GPT-4 consistently consider longer answers to be more helpful. Similar to the discussion in Section 4.2, we believe that this preference stems from the strong correlation between "helpfulness" and "the level of detail", as longer answers tend to convey more information, making them perceived as more helpful. Regarding the "Language" dimension, recognizing spelling or grammatical errors in text is challenging for experts, while GPT-4 effectively distinguishes between answers based on their language quality and appears to penalize grammatical errors more heavily during assessment. Overall, this comprehensive analysis sheds light on the evaluation process and reveals the differing perspectives of expert annotators and GPT-4 in assessing various dimensions of model performance.

## 5 DISCUSSION

When evaluating the quality of an LLM and selecting the best model, it is important to consider various factors, including accuracy, fluency, and other relevant aspects. However, combining these factors into a single score is not recommended, as it is unclear which factor should be given the highest priority for both the LLM itself and human judges. Therefore, it is crucial to analyze and evaluate these factors individually for a comprehensive assessment. We propose breaking them into three categories. However, these categories may not be the perfect setup to capture all the aspects required for an ideal answer, so further research is necessary.

Another important consideration is the use of human judgments for evaluating LLM performance. While crowd feedback can provide a general indication of how the audience perceives the LMs' output, caution must be exercised. Crowd-sourced evaluators may not always involve rigorous factchecking, thus giving high scores to factually incorrect answers. Expert evaluators are better in this aspect, with the caveat of more difficulty in scaling the process. Additionally, both human annotators demonstrate various biases, such as the length and order of the answers. Hence, it is crucial to supplement human opinions with other evaluation methods to gain a more comprehensive understanding of the LMs' capabilities.

## 6 LIMITATIONS

**Question Coverage**   We select only 40 questions from Chiang et al. (2023). We acknowledge that this limited selection may not capture the full spectrum of question types and variations. Consequently, there is a potential risk that some aspects of the research question may not receive sufficient representation or exploration.

**Evaluation Dimension Coverage**   In our proposed Multi-Elo Rating System, we only explore three crucial dimensions: "Accuracy", "Helpfulness", and "Language". We acknowledge that while these dimensions provide valuable insights, they may not encompass the entirety of the multifaceted

nature of text evaluation. Furthermore, it is important to recognize that our definitions for the three dimensions we have chosen are not infallible. Different stakeholders may have diverse perspectives on these dimensions.

We leave the investigation of addressing these limitations to future work.

## 7 RELATED WORK

**Large Language Models**    Large Language Models (LLMs) commonly refer to Transformer-based language models with billions of parameters (Vaswani et al., 2017). Examples of these models include GPT-3 (Brown et al., 2020), PanGu-$\alpha$ (Zeng et al., 2021), Chinchilla (Hoffmann et al., 2022), PaLM (Chowdhery et al., 2022), BLOOM (Scao et al., 2022), Galactica (Taylor et al., 2022), and LLaMA (Touvron et al., 2023). These models, trained on massive datasets, demonstrate impressive abilities in understanding natural language and handling complex tasks. It is found that instruction-tuned LLMs can significantly enhance their performance following general language instructions (Weller et al., 2020; Mishra et al., 2022; Wang et al., 2022b; Wei et al., 2022; Sanh et al., 2022; Ouyang et al., 2022; Parmar et al., 2022; Scialom et al., 2022; Chung et al., 2022; Yin et al., 2022; Gupta et al., 2022; Muennighoff et al., 2022). Therefore, accurately and comprehensively assessing the performance of these LLMs remains an unsolved challenge. This study aims to examine the effectiveness of both humans and LLMs as evaluators.

**Evaluation**    With the advancement of LLMs (Large Language Models), the need for their thorough evaluation becomes increasingly important. Traditionally, NLP models are assessed using standardized benchmark test suites. Given the capabilities of LLMs, several studies suggest using a diverse set of NLP benchmarks for a more comprehensive understanding (Hendrycks et al., 2021; Gao et al., 2021; Srivastava et al., 2022; Liang et al., 2022; Li et al., 2023b). As pointed out by Gudibande et al. (2023) and Zheng et al. (2023) that there is a gap between users' perception and standardized evaluation suites, recent LLM studies often incorporate human evaluation for a more nuanced understanding of model performance (Wang et al., 2022a; Chiang et al., 2023; Wu et al., 2023). As human evaluations can be costly, some recent works utilize state-of-the-art LLMs such as GPT-4 (OpenAI, 2023) and Claude-1 (Bai et al., 2022b) to evaluate model outputs. More recently, several works employ the Elo rating system from chess games to gauge the LLMs' capabilities (Askell et al., 2021; Bai et al., 2022a; Srivastava et al., 2022; Dettmers et al., 2023; Zheng et al., 2023). However, these previous works operate under the assumption that human evaluations serve as the gold standard. In contrast, Clark et al. (2021) argue that human evaluation is no longer the gold standard, highlighting the inability of human judges to distinguish between human-written and machine-generated text. Their research focuses solely on language quality, while our investigation extends to evaluating the behavior of both humans and LLMs across three aspects: factual accuracy, language quality, and text length.

## 8 CONCLUSION

In this study, we investigate the limitations of human judges and LLMs as evaluators by examining their behaviors in assessing machine-generated text. We deliberately introduce fabricated factual and grammatical errors into a set of machine-generated answers and analyze the responses of crowd-sourced, expert, and LLM judges. The primary goal is to gain insights into the limitations and biases exhibited by both human and LLM judges. We observe that both human judges and LLMs demonstrate various biases.

To address the observed issues, we propose to assesses machine-generated text across multiple dimensions independently and instantiate such an idea with the Elo rating system, leading to the Multi-Elo Rating System. Our empirical findings demonstrate the effectiveness of this approach in enhancing the evaluation quality of GPT-4, particularly in terms of factual accuracy. However, crowd judges continue to display indecisiveness. Given these findings, we encourage practitioners to adopt a multi-dimensional approach when evaluating machine-generated text, rather than relying solely on a single unified measure. We also recommend caution when using crowd annotators to assess LLMs due to their indecisiveness, bias toward lengthy responses, and their limited capability in fact-checking answers.

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

```
Question/Instruction:
$instruction

Answer the question/instruction.
The answer should be roughly 100 words long.
```

Figure 3: The prompt employed by GPT-4 for generating answers of the "`Correct`" model.

```
Question/Instruction:
...

Answer the question/instruction.
The answer should be roughly 100 words long.
The answer must contain one minor factual error.
The factual error can be made-up names, wrong numbers, incorrect facts,
↪  or incorrect suggestions.
List the error and its corresponding justification separately.
Enclose your answer within <answer> and </answer> tags.
Enclose the error and justification within <error> and </error> tags.
```

Figure 4: The prompt employed by GPT-4 for generating answers of the "`One Minor Factual Error`" model.

## A    ANSWER GENERATION PROMPTS

We present the answer generation prompt for "`Correct`" (Figure 3), "`One Minor Factual Error`" (Figure 4), "`Several Minor Factual Errors`" (Figure 5), "`Several Major Factual Errors`" (Figure 6), "`Advanced Learner`" (Figure 7), and "`Intermediate Learner`" (Figure 8) in this section. For those prompts generating short answers, we simply require GPT-4 to ensure that "The answer should be roughly 50 words long".

## B    LLM EVALUATION PROMPT

We utilize the evaluation prompt for LLMs from Dettmers et al. (2023), as presented in Figure 9.

## C    HUMAN EVALUATION INTERFACE

The interface used for crowd-source evaluation is shown in Figure 10 and Figure 11

```
Question/Instruction:
...

Answer the question/instruction.
The answer should be roughly 100 words long.
The answer must contain several minor factual errors.
The factual errors can be made-up names, wrong numbers, incorrect
↪  facts, or incorrect suggestions.
List the errors and their corresponding justifications separately.
Enclose your answer within <answer> and </answer> tags.
Enclose the errors and justifications within <error> and </error> tags.
```

Figure 5: The prompt employed by GPT-4 for generating answers of the "`Several Minor Factual Errors`" model.

```
Question/Instruction:
...

Answer the question/instruction.
The answer should be roughly 100 words long.
The answer must contain several major factual errors.
The factual errors can be made-up names, wrong numbers, incorrect
↪  facts, or incorrect suggestions.
List the errors and their corresponding justifications separately.
Enclose your answer within <answer> and </answer> tags.
Enclose the errors and justifications within <error> and </error> tags.
```

Figure 6: The prompt employed by GPT-4 for generating answers of the "Several Major Factual Errors" model.

```
Question/Instruction:
...

Answer the question/instruction.
The answer must be written as if you're an advanced-level English
↪  learner.
The answer must contain 2 or 3 minor grammatical and spelling errors.
The answer should be roughly 100 words long.
List the errors and their corresponding justifications separately.
Enclose your answer within <answer> and </answer> tags.
Enclose the errors and justifications within <error> and </error> tags.
```

Figure 7: The prompt employed by GPT-4 for generating answers of the "Advanced Learner" model.

```
Question/Instruction:
...

Answer the question/instruction.
The answer must be written as if you're an intermediate-level English
↪  learner.
The answer must contain 5 or more major grammatical and fluency errors.
The answer must be roughly 100 words long.
List the errors and their corresponding justifications separately.
Enclose your answer within <answer> and </answer> tags.
Enclose the errors and justifications within <error> and </error> tags.
```

Figure 8: The prompt employed by GPT-4 for generating answers of the "Intermediate Learner" model.

```
[Question]
...

[The Start of Assistant 1's Answer]
...
[The End of Assistant 1's Answer]

[The Start of Assistant 2's Answer]
...
[The End of Assistant 2's Answer]

[System]
We would like to request your feedback on the performance of two AI
↪  assistants in response to the user question displayed above.
Please rate the helpfulness, relevance, accuracy, level of details of
↪  their responses. First, provide your evaluation of the assistant's
↪  helpfulness, relevance, accuracy, and level of detail. Please
↪  provide a comprehensive explanation of your evaluation, avoiding
↪  any potential bias and ensuring that the order in which the
↪  responses were presented does not affect your judgment.
Once you have carefully reviewed both submissions, in a new line,
↪  choose between the answers of Assistant 1 and Assistant 2 by
↪  outputting the number 1 or 2 respectively, or choose 3 if the two
↪  assistants are equivalent. Do not output anything else other than
↪  the number in this last line.
```

Figure 9: The evaluation prompt employed by GPT-4 and Claude-1 is designed to assess the quality of responses. These language models evaluate answers based on criteria such as helpfulness, relevance, accuracy, and level of detail.

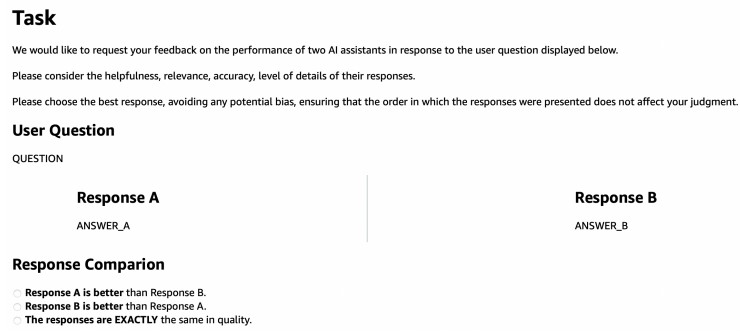

Figure 10: Annotation interface for single Elo score.

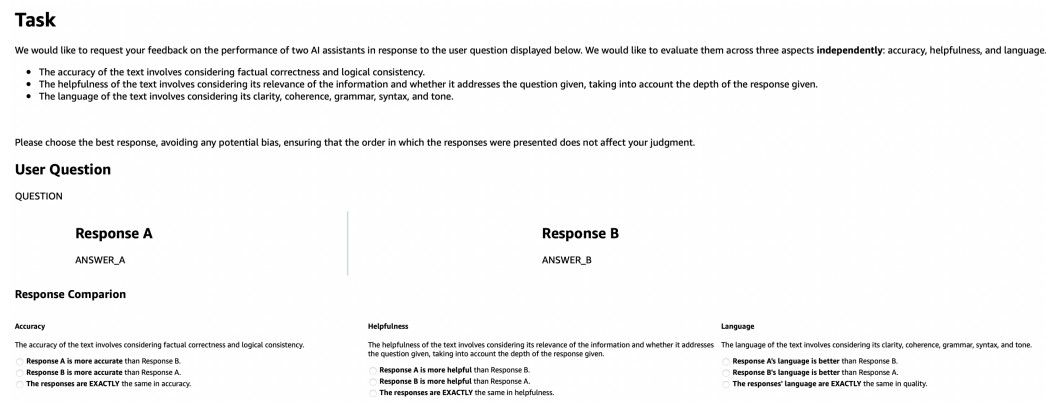

Figure 11: Annotation interface for multiple Elo scores.

| | Accuracy | | Helpfulness | | Language | |
|---|---|---|---|---|---|---|
| | Separate | Compound | Separate | Compound | Separate | Compound |
| Correct | 1200 | 1284 | 1384 | 1429 | 1415 | 1429 |
| + Short | 1158 | 1146 | 1009 | 1054 | 1199 | 1178 |
| One Minor Factual Error | 1120 | 1221 | 1378 | 1399 | 1334 | 1346 |
| + Short | 1016 | 1045 | 965 | 993 | 1109 | 1090 |
| Several Minor Factual Errors | 993 | 1054 | 1248 | 1208 | 1200 | 1187 |
| + Short | 857 | 895 | 845 | 833 | 988 | 956 |
| Several Major Factual Errors | 794 | 805 | 926 | 884 | 995 | 968 |
| + Short | 746 | 730 | 726 | 711 | 871 | 842 |
| Advanced Learner | 1139 | 1178 | 1196 | 1210 | 1051 | 1093 |
| + Short | 1051 | 969 | 804 | 810 | 814 | 839 |
| Intermediate Learner | 1018 | 911 | 908 | 853 | 560 | 565 |
| + Short | 908 | 761 | 612 | 615 | 465 | 506 |

Table 5: The Elo ratings for different models with regard to "Accuracy", "Helpfulness", and "Language" given by GPT-4. "Separate" means that GPT-4 assesses the factual accuracy of the models using a separate prompt, while "Compound" implies that GPT-4 evaluates all three dimensions simultaneously using a compound prompt.

## D  SEPARATE VS. COMPOUND

In this section, we explore two evaluation strategies: assessing each dimension separately or evaluating all dimensions simultaneously using a compound prompt. The results obtained from GPT-4 using these two strategies are presented in Table 5.

Regarding the "Accuracy" dimension, our findings indicate that GPT-4 performs better when assessing the factual accuracy of models independently. However, when using the compound prompt, GPT-4 ranks "One Minor Factual Error" and "Advanced Learner" higher than "Correct + Short". This observation leads us to hypothesize that evaluation dimensions can mutually influence each other when evaluated concurrently, even when explicitly instructing GPT-4 to evaluate each dimension independently. For the "Helpfulness" dimension, GPT-4, when using the separate prompt, ranks "Several Minor Factual Errors" higher than "Advanced Learner". However, when using the compound prompt, GPT-4 ranks "Advanced Learner" higher than "Several Minor Factual Errors". Interestingly, the "Language" dimension is the most consistent, as GPT-4 produces the same rankings using both evaluation strategies.

Based on our findings, we choose to use the separate prompts for each dimension in this work, as this strategy yields better results in terms of factual accuracy.

