# OpenReview forum: "Style Over Substance: Evaluation Biases for Large Language Models"
_ICLR.cc/2024/Conference — Submitted to ICLR 2024_

### Official Review · Reviewer_X9cm · 2023-11-01

**Soundness:** 2 fair
**Presentation:** 2 fair
**Contribution:** 2 fair
**Rating:** 3
**Confidence:** 4

**Summary:**

This paper studies weaknesses with using LLMs as proxies for human judges. To do so, the authors take 40 questions, then have GPT-4 generate responses that satisfy different personas (e.g., an advanced English learner versus an intermediate English learner), then modify the responses when GPT-4 does not satisfy the specified criteria. The authors then use an ELO analysis and finds that writing with “several factual errors” outperforms “correct + short”, while finding that order impacts judge’s decisions, and humans tend to be inconsistent. To evaluate better, the authors introduce MERS, which evaluates responses ion terms of “Accuracy”, “helpfulness” and “Language”, and find that the ratings of these subcategories more closely align with the ways the different promptings of GPT-4 can be ranked.

**Strengths:**

* The work studies an important problem: assessing the reliability of ELO style evaluation, that is increasingly adopted
* The decomposition trick (i.e., evaluating multiple attributes instead of a single preference) seems general and useful.

**Weaknesses:**

* The study is small scale: it only analyzes forty questions, which makes it hard to make systematic conclusions.
* The study only considers three different attributes to evaluate, and only a few types of generated answers. It’s possible these three attributes are useful for identifying problems with these types of answers, but not others
* The characteristics rely on GPT-4 faithfully following the prompt, which doesn’t work, so the authors manually edit the text. This adds a human bias to the study, which could impact the results.

**Questions:**

Do you expect the results to change with a larger study, or with different attributes (e.g., beyond "helpfulness") that you measure?

---

> ### Author Response · Authors · 2023-11-15
>
> Thank you for your review and constructive feedback on our manuscript.
>
> 1. Study scale
>
> The reviewer highlights concerns regarding the small scale of our study, which involved only 40 questions. However, it is important to note that our analysis extended far beyond the raw number of questions. The evaluation encompassed these questions across eight different models, resulting in **5280 individual comparisons**. The total resource and financial expenditure for this scale of analysis amounted to **thousands of USD.**
>
> **Moreover, our analysis demonstrates a consistent trend in the Elo rankings when sampling from these 40 questions.** This consistency indicates that, despite the seemingly small sample size, the dataset is robust enough to yield reliable insights and trends. Therefore, we believe that the chosen sample size and methodology strike a delicate yet effective balance between comprehensive analysis and practical constraints such as cost and resource allocation.
>
> 2. Limited evaluation attributes
>
> Our study focused on three attributes – accuracy, helpfulness, and response length – based on their relevance and importance in the context of LLM evaluation. We agree that exploring additional attributes could provide a more comprehensive understanding of LLM capabilities. **This choice was a strategic decision to delve deeper into a few significant aspects rather than superficially covering a broader range.** Future research could incorporate more diverse attributes, such as creativity, engagement, or cultural sensitivity, to enrich the evaluation framework.
>
> 3. Reliance on manual editing:
>
> **This step was necessary to ensure that the responses adhered to the study's requirements**, as GPT-4 did not always generate outputs that aligned with the intended error types or language proficiency levels. We would like to highlight that only a few examples required post-editing.
>
> 4. Larger study scales or more attributes
>
> Regarding the question about the potential impact of a larger study or different attributes, we believe that expanding the study scale and incorporating varied attributes would likely yield more nuanced insights and possibly reveal additional biases or evaluation trends. However, **the core findings regarding the preference for longer, factually incorrect responses over shorter, accurate ones still hold,** as these reflect deeper biases in human and LLM evaluation processes. A larger-scale study would help in validating these findings and understanding their generalizability.

---

### Official Review · Reviewer_fkGz · 2023-11-10

**Soundness:** 3 good
**Presentation:** 2 fair
**Contribution:** 3 good
**Rating:** 5
**Confidence:** 4

**Summary:**

The paper investigates evaluation of LLM-generated answers by human experts, crowd evaluators as well as LLMs.
The GPT-4 model is used to generate the answers to 40 questions, and different instructions/prompts were used to generate different types of outputs (with major factual errors, with minor factual errors, with grammar errors).
GPT-4 is also used for evaluation, but one more LLM (Claude-1) is also used in order to avoid potential biases of the model towards its own output.

The outputs are then ranked pairwise and Elo rating (designed for chess) is computed from all rankings.
In addition, apart from using a single overall criterion for ranking, using three different aspects separately is presented, too: accuracy, helpfulness and language.

The main findings are that in all evaluations longer answers are preferred, that crowd evalutors often disregard factual errors, that human annotators often disregard grammar and spelling errors.

**Strengths:**

Evaluating LLM outputs is important, so it is great to see exploring the evaluation process by different evaluators.
Since LLMs are more and more used for evaluating NLP tasks, it is also important to analyse and understand their capabilities and flaws.

Considering different quality aspects is great, too.

**Weaknesses:**

Several important parts are not clearly presented or defined.
Putting related work at the end also makes the reading harder -- for example, it was not clear until the end if Elo rating is a widely used approach.
 Also, it is not clear whether  the mentioned criteria are usually used for evaluating question answering, or it is something proposed in this work. It seems that evaluating on different critera separately is new.

It is not clear why only GPT-4 is used for the evaluation on different criteria, and not the other LLM (Claude-1) -- this is important, especially because of the bias toward own outputs (mentioned by authors, too).

Some terms should be revised for the sake of clarity (e.g. replace "games" by "pairwise comparisons", "answer order" by "order of outputs", etc.

More details can be found in "Questions".

**Questions:**

Introduction

In this study, we systematically generate a set of responses
=> not clear if evaluation responses were generated, or texts to be evaluated


we observe hesitancy among human
=> what is "hesitancy" in this context?

 while LLMs lean toward the first answer and expert annotators prefer the second answer
=> what are "the first answer" and "the second answer"?

Overall, the entire paragraph starting with "In the course of our investigation (Table 1 and Section 3)" is much more suitable for Discussion than for Introduction.


It is not clear what are the widely used evaluation methods and criteria: for example, is calculating Elo rating from pairwise rankings widely used or only in some publications?


Section 2

Section 2.2 "answer generation" is out of place within the section about the evaluation, it should be  a separate section (Section 2, also with corpus statistics), and then Section 3 only evaluation.

Is the set of questions publicly available? If yes, on which link?

Why not explicitly ask for spelling errors and grammatical erros, instead of giving learning level and supposing the type of errors?

answer ordering
=> does this refer to order of outputs of different mdoels (set-ups)?
then it would be better to call it "order of generated outputs"

Section 2.3
The guidelines for the annotators are very important and should be part of the main sections, not only in Appendix.

Section 2.4
we sample 200 games out of a total pool of 5280 games
annotate 10 games
=> which games?

This approach enables us to accurately evaluate the effectiveness and reliability of each annotation method.
=> Why/How?

Section 2.5
studies relies => studies rely

which may not be appropriate for our work as our answers are refined by humans after being generated by GPT-4
=> why refining by humans can be inappropriate?
using the same LLM for generating outputs and for evaluating them can indeed be inappropriate

The prompt assesses the answers based on their helpfulness, relevance, accuracy, and level of detail
=> were the same instructions/guidelines given to human annotators?
The guidelines for the annotators should be mentioned/explained in the previous sub-sections.

footnote 2) it seems that "games" refer to pairwise comparisons?
the term "games" is confusing, it should be replaced by "comparisons"


Section 3

As anticipated, the standard correct model attains the highest Elo score across all human
judges. Nevertheless, intriguing disparities emerge when comparing the Elo results from human
judges to those generated by GPT-4 and Claude-1.
=> Not clear: the standard correct model also has highest scores according to GPT-4 and Claude-1? What are the intriguing disparities?


Figure 2:
What are "Assistant 1" and "Assistant 2"?
there is no explanation in the table caption

 leading to Elo scores that remain relatively close to the initial
value of 1000 +
"by assigning a high number of ties"

something like this should be added in order to make the statements fully clear

Why is this necessarily a disadvantage?
Especially if the outputs are similar in the important aspects?

"the order of answers" is the order of generated outputs, right?
What is "Assistant 1" and "Assistant 2"?
What is exactly the order of the answers? Which is the first output and which is the second?
They should be presented at the same time in order to be ranked?
Is the first on the left and the second on the right?
Or the first is on above the second?
Or something else?

those model => those models

particularly when they appear convincing => what "convincing" means exactly?
What is the difference between human failures of fact-checking and LLMs' failures of fact-checking?
Or they are of the same nature?
Why there is no figure with percentages? (similar to Figure 1 and Figure 2)

What values were used for calculating Kappa coefficient?
Because the evaluation consists of rankings, not of assigning numbers.
Why not for example Kendal's Tau?


Section 4

the current widely used evaluation approach => what is that exactly?
Calculating Elo rating from pairwise rankings?
Guidelines/instructions for ranking (what to pay attention to:  accuracy, level of details, relevance, language, helpfulness


Recent advancements in MT evaluation also advocate breaking down the evaluation into multiple aspects
=> this is error annotation, not ranking

MQM is error annotation framework, not really related to different overall quality aspects
in this submission, there is absolutely no error annotation, only pairwise ranking
an appropriate analogy to MT could be separate ranking according to adequacy, fluency, readability

Table 4
Why Claude-1 is not used for this experiment (evaluating separately according to three aspects)?


Section 5
it is crucial to supplement human opinions with other evaluation methods => are "other evaluation methods" using LLMs for evaluation?


Related work should follow Introduction, not be placed almost at the end of the paper.

---

> ### Author Response · Authors · 2023-11-15
>
> Thank you for your detailed and constructive feedback on our manuscript.
>
> 1. Clarity and presentation of concepts
>
> You pointed out that several parts are not clearly presented or defined. We acknowledge this concern and will revise the manuscript to enhance clarity including the placement of the related work, the human annotation guidelines, and so on. We will also clearly define the terms, like “game” and “hesitancy”, in the future revisions of our manuscript
>
> 2. Elo ratings
>
> Elo ratings are a method for calculating the relative skill levels of players in two-player games such as chess. In each game, the winner gains points from the loser. **Elo ratings have been adopted to evaluate Language Learning Models (LLMs) in recent publications (Bai et al., 2022; Dettmers et al., 2023; Zheng et al., 2023) and a popular LLM leaderboard (https://arena.lmsys.org/).**
>
> In this work, we utilize crowd-sourced annotators, expert annotators, GPT-4, and Claude-1 as evaluators to perform pairwise comparisons. Each comparison involves one question and two answers provided by two different models, and we ask the evaluators to determine which answer is of better quality. The prompt used for LLM evaluation is presented in Figure 9, and the human annotation guideline is in Figure 10. Assistant 1 and Assistant 2 are the two models used in each comparison.
>
> References:
>    - Bai, Yuntao, et al. "Training a helpful and harmless assistant with reinforcement learning from human feedback." arXiv preprint arXiv:2204.05862 (2022).
>    - Dettmers, Tim, et al. "Qlora: Efficient finetuning of quantized llms." arXiv preprint arXiv:2305.14314 (2023).
>    - Zheng, Lianmin, et al. "Judging LLM-as-a-judge with MT-Bench and Chatbot Arena." arXiv preprint arXiv:2306.05685 (2023).
>
> 3. MQM
>
> **Our Multi-Elo Rating System is inspired by MQM, though it is not identical to MQM.** The concept of breaking down the evaluation into several categories effectively improves the interpretation and transparency of the evaluation results.
>
> 4. Lack of Claude
>
> **Due to cost constraints and the results in Table 1, we did not include Claude-1 in Table 4.** As demonstrated in Table 1, Claude-1 shows similar results to GPT-4, therefore, we used only GPT-4 in subsequent experiments.
>
> 5. Question set
>
> The questions used in this work are available at https://github.com/lm-sys/vicuna-blog-eval/blob/main/eval/table/question.jsonl

---

### Official Review · Reviewer_2ems · 2023-11-11

**Soundness:** 1 poor
**Presentation:** 2 fair
**Contribution:** 2 fair
**Rating:** 3
**Confidence:** 5

**Summary:**

The paper is focusing on the important open problem of evaluation in natural language generation. Using a dataset consisting of 40 questions of intentionally flawed machine generated (GPT4) answers, the authors employ both human evaluation (crowd-sourced and expert) and LLM-based evaluation focusing on three aspects of the generated texts: i) language proficiency, ii) factual accuracy, and iii) response length. Their analysis reveals biases in the evaluation process: human hesitancy vs LLMs certainty in determining answer quality, humans do not thoroughly fact-check answers vs LLMs displaying some degree of fact-checking ability, and both human and LLM judges favor longer texts. Moreover, factually incorrect models are preferred over those that generate grammatically incorrect or short responses. Informed by these findings, the authors recommend the evaluation of texts from multiple perspectives as opposed to a single metric only, and propose a rating system that assesses the quality of machine-generated texts accounting for i) language, ii) accuracy, and iii) helpfulness criteria.

**Strengths:**

The topic of the paper is very relevant and timely since evaluation is an open problem in the literature. The community needs a thorough understanding of current evaluation metrics and informed by their limitations,  better evaluation metrics. The paper makes a step in this direction.

**Weaknesses:**

Their evaluation setup is limited and includes dataset of only 40 questions. The sample size is small and insufficient for drawing convincing conclusions. The authors also do not discuss the importance of the categories of these questions (“generic”, “knowledge”, “common sense”, and “counterfactual”) after leaving out other categories.


The expert evaluation section (2.4) discusses games, but these games are not introduced/presented ahead of this section: “we sample 200 games out of a total pool of 5280 games, which is statistically sufficient to yield meaningful in- sights into the model’s performance”.

“By incorporating multiple LLM judges, we can establish a more comprehensive and unbiased assessment of the generated answers in our study.” - Generally agree with this statement, however in the experiments only GPT-4 and Claude (Antrophic) are used, how is this in line with your claims? Only two LLM-based models is not reflective of the goals of your statement. In addition, Claude is not even mentioned in the Introduction or anytime before Section 2.5, leaving the reader the impression you are only using GPT4.

The findings of the user study conducted by the authors are unsurprising. The length of the text is a well-known confounder in the evaluation of generated texts. Similarly, texts which contain grammatical errors can be quickly identified as having poor quality. Moreover, it is unsurprising that humans tend to be biased towards factually incorrect answers that look coherent and are grammatically correct (especially true if they lack in-domain knowledge). Besides, the authors make misleading statements when claiming LLMs reach consensus when in fact GPT-4 and Claude only reach moderate agreement, and that LLMs display some degree of fact checking abilities - hard to quantify this statement, since the lack of factuality is an open problem with current models. Similarly, the authors claim “the expert annotators outperform the crowd-sourced annotators in the evaluation, despite not being entirely error-free themselves.” - this is a well known fact in the literature, however it is presented as if it were a novel insight which is very misleading!

“expert annotators indeed prioritize factual accuracy during evaluation.” -  of course, since they have in-domain knowledge. However, there is no discussion highlighting the relevance of the selected experts vs the selection of the specific domains the authors choose to focus on in this work.

“Regarding the “Helpfulness” dimension, both expert annotators and GPT-4 consistently consider longer answers to be more helpful.” - of course, since they contain more information! Nevertheless, the authors only make vague assumptions and fail to quantify the degree of factuality to support their statements.


The setup is not consistent: it evaluates i) language proficiency, ii) factual accuracy, and iii) response length, and then proposes a rating system that accounts for  i) language, ii) accuracy, and iii) helpfulness. Helpfulness criteria is not clearly motivated, and its inclusion is inconsistent with the initial evaluation criteria. Besides, the paper does not clearly describe how this aspect is being evaluated, to whom it is helpful and in which contexts.



There is no clear motivation in the paper on why the authors chose to focus on these particular three aspects ( language proficiency, factual accuracy, and response length) of machine-generated texts. There are so many aspects of the generated texts that can be considered, and this paper fails to make a convincing argument on the selection of their evaluation criteria. While the authors acknowledge in the Discussion section that this may not be the perfect setup, they fail to motivate their experimental choice.

The paper fails to discuss limitations of human evaluation, as well as limitations of LLM-based evaluation with respect to findings already well-established in the literature. Their statements in the Discussion section are vague and unsupported by evidence.

The Related Work section is not very informative for the topic of the paper, particularly the paragraph focused on enumerating existing LLMs is not discussing evaluation at all (which is the main focus of the current paper). In addition, the authors fail to cite and refer to important bodies of work in the literature focused on both human and machine evaluation. I recommend a complete rewrite of this section and instead of presenting generalities, discuss papers that have addressed the same topic and are particularly relevant to your current work.

In general, the experiment setup decisions made in this paper seem rather arbitrary and not well documented / supported by the arguments presented. The authors fail to discuss and include significant papers focused on the evaluation of natural language generation (NLG) models, and do not place their work in the context of recent literature on NLG evaluation.

**Questions:**

This sentence is confusing: “we limit each annotator to provide no more than 20 annotations and enforce a strict maximum of 50 annotations per person.” - Does an annotator judge 20 or 50 questions?

How do you evaluate helpfulness?

---

> ### Author Response · Authors · 2023-11-15
>
> Thank you for your detailed and constructive feedback on our manuscript.
>
> 1. Sample size and questions
>
> Regarding our dataset size of 40 questions, it is crucial to highlight the significant resource and cost implications involved in our study. The evaluation of these questions against twelve different models has led to **5280 individual comparisons**, with the total expenditure amounting to **thousands of USD**. This scale of analysis, while seemingly compact, is both resource-intensive and costly.
>
> Moreover, **our analysis demonstrates a consistent trend in the Elo rankings when sampling from these 40 questions**. This consistency indicates that the dataset is robust enough to yield reliable insights and trends. Therefore, we believe that the chosen sample size strikes an effective balance between comprehensive analysis and practical constraints.
>
> 2. "Games"
>
> **Our use of the term "games" refers to pairwise comparisons of model outputs**, similar to chess matches in the Elo rating system. Each expert annotator evaluated 10 such comparisons. We will update our manuscript as you suggested.
>
> 3. LLM Judges and Model Diversity
>
> **By definition, “multiple” means “more than one”, and our claim remains consistent with our methodology.** We incorporated Claude-1 alongside GPT-4 to counter potential biases, especially relevant since our answers were refined post-generation by GPT-4. The inclusion of more LLMs was considered; however, we had concerns about their capabilities in accurately evaluating model outputs, coupled with limited resources for this study. This decision was a strategic choice to maintain focus and ensure quality in our evaluation process.
>
> 4. Unsurprising findings
>
> We believe that the value of our study lies not only in uncovering new insights but also in empirically validating existing assumptions within the field. While some of our findings may align with general expectations, their empirical substantiation is a crucial contribution. This validation is particularly significant in a rapidly evolving field like natural language processing, where many widely-held beliefs have yet to be rigorously tested. Our study offers concrete evidence to support or question these beliefs, thus advancing the overall understanding of natural language generation evaluation. **We believe that unsurprising findings should not be a reason for rejection, as the single-score Elo rating is still widely adopted, and general expectations require rigorous scientific confirmation.**
>
> Furthermore, the study's design and the nuances of our findings offer deeper insights into the complexities of human and machine evaluations, going beyond mere confirmation of expected trends. For example, there are few grammatical errors in your review which are hard to be identified without the assistance of the grammar-checker. This actually confirms our claim that humans are not able to effectively identify the errors in the text. In summary, while some findings might seem predictable, their empirical confirmation and the detailed analysis provided are valuable contributions to the field.
>
> 5. Setup inconsistencies
>
> The initial phase of our evaluation concentrated on three key aspects: language proficiency, factual accuracy, and response length. These were selected to rigorously assess the biases of the LLM evaluation process. In transitioning to our Multi-Elo Rating System (MERS), we selected aspects i) language, ii) accuracy, and iii) helpfulness. This selection was made for demonstrative purposes, to showcase the system's capability in evaluating diverse and crucial elements of language generation. **It is important to note that our proposed MERS is adaptable and not limited to these three aspects.** The system is designed to be flexible, allowing for the inclusion of additional evaluation criteria as needed.
>
> 6. Related Work
>
> In the related work, we present numerous researches that are related to our study, **covering the literature of both large language models and evaluation methods.** We will include more specific references to works directly addressing NLG evaluation in our future revisions.
>
> 7. Annotation workload
>
> To ensure the annotation diversity and quality, we limit the annotation workload of each crowd-sourced annotator. **We recommend them to annotate no more than 20 samples.** Given that AMTurk doesn't have a built-in feature to enforce this limit, it serves as a general guideline for annotators. However, in practice, we do accommodate instances where annotators exceed this limit. Annotators can submit more than 20 annotations, but we closely monitor their contributions. **We choose to disregard submissions from any annotator who exceeds 50 annotations.**
>
> 8. Helpfulness
>
> As shown in Figure 11, we require the human annotators, including the crowd-sourced annotators and experts, to **consider the relevance of the information, whether it addresses the given question, and to take into account the depth of the response.**

---

### Meta-Review · Area_Chair_S6Nf · 2023-12-13

**Metareview:**

The paper investigates evaluation of LLM-generated answers by human experts, crowd evaluators and LLMs. GPT-4 is used to generate answers for 40 questions, and different instructions/prompts were used to generate different types of outputs. GPT-4 and Claude-1 are used as LLM evaluators.
Overall, all reviewers agree that the topic is relevant and important. It is also great that the authors look at different quality aspects.
However, there are a couple of concerns. First of all, the study only includes 40 questions and it is unclear how stable the conclusions are. Another concern is the generalization of this study. All outputs are generated by the same model (GPT-4) and it is unclear if this covers all possible corner cases. Another concern is that some output needs human edits and might bias the output. Further, it is unclear if looking at three quality aspects generalizes. The authors acknowledge the concern, but also raise the concern that including more questions would be costly.
There is also some concern regarding the completeness of the related work, especially regarding the human evaluation background. Reviewers would like to see more clarity why specific design decisions (e.g. which aspects to focus on) were chosen and get a more clean description of helpfulness.

**Justification For Why Not Higher Score:**

All reviewers raised valid concerns and the current version of the paper should not be accepted.

**Justification For Why Not Lower Score:**

N/A

---

### Decision · Program_Chairs · 2024-01-16

Reject